# Hemozoin-catalyzed precipitation polymerization as an assay for malaria diagnosis

Omar Rifaie-Graham[1], Jonas Pollard [1], Samuel Raccio [1], Sandor Balog [1], Sebastian Rusch[2,3], María Andrea Hernández-Castañeda[4], Pierre-Yves Mantel[4], Hans-Peter Beck [2,3] & Nico Bruns [1]

Methods to diagnose malaria are of paramount interest to eradicate the disease. Current methods have severe limitations, as they are either costly or not sensitive enough to detect low levels of parasitemia. Here we report an ultrasensitive, yet low-resource chemical assay for the detection and quantification of hemozoin, a biomarker of all *Plasmodium* species. Solubilized hemozoin catalyzes the atom transfer radical polymerization of *N*-isopropylacrylamide above the lower critical solution temperature of poly(*N*-isopropylacrylamide). The solution becomes turbid, which can be observed by naked eye and quantified by UV-visible spectroscopy. The rate of turbidity increase is proportional to the concentration of hemozoin, with a detection limit of 0.85 ng mL$^{-1}$. Malaria parasites in human blood can be detected down to 10 infected red blood cells µL$^{-1}$. The assay could potentially be applied as a point-of-care test. The signal-amplification of an analyte by biocatalytic precipitation polymerization represents a powerful approach in biosensing.

[1] Adolphe Merkle Institute, University of Fribourg, Chemin des Verdiers 4, 1700 Fribourg, Switzerland. [2] Swiss Tropical and Public Health Institute, Socinstrasse 57, 4051 Basel, Switzerland. [3] University of Basel, Petersgraben 4000 Basel, Switzerland. [4] Department of Medicine, University of Fribourg, Route Albert-Gockel 1, 1700 Fribourg, Switzerland. These authors contributed equally: Omar Rifaie-Graham, Jonas Pollard, Samuel Raccio. Correspondence and requests for materials should be addressed to N.B. (email: nico.bruns@unifr.ch)

Malaria is a common and life-threatening parasitic disease caused by several *Plasmodium* species that is transmitted via mosquito bites. In 2017, the WHO (World Health Organization) estimated 3.1 billion people at risk of being infected by malaria, with 1.1 billion people at high risk. 219 million cases, an increase of 3 million cases over the previous year, were reported ending in 435,000 deaths[1]. Africa remained as the region with the highest mortality burden accounting for 93% of the deaths. Sixty-one percent of the worldwide deaths were children under the age of five. Detection of malaria infections is one of the keys to eradication of the disease. For example, highly sensitive techniques to screen populations in endemic areas are needed to detect and subsequently treat asymptomatic individuals with submicroscopic *Plasmodium sp.* infections who do not display symptoms, but who represent a major reservoir for the transmission of the disease[2,3]. An effective field test for malaria must be highly sensitive, its reagents have to be robust to withstand transport and storage in tropical regions, it has to be easy to use and should be of low cost. Today, no single malaria field test on the market fulfills all these criteria. Microscopy in combination with Giemsa staining is the most widely used technique to diagnose malaria[4]. It allows the identification of the malaria species and has a detection threshold of ~50 infected red blood cells (iRBC) $\mu L^{-1}$ [5]. However, this method requires well-trained microscopists to correctly identify the parasite in a blood smear. As a result, this diagnosis is mostly performed at central laboratories or hospitals. This is the reason why malaria rapid diagnostic tests (MRDTs) were developed for field use. Since the WHO recommended in 2010 the diagnosis of all suspected cases, the number of people using MRDTs has increased from <200,000 in 2005 to more than 314 million in 2014[6,7]. There are more than 200 MRDTs currently available;[8] most of them are based on immuno-chromatography strips which perform antigen–antibody complexation[8]. However, their sensitivity to malarial parasites is relatively low (detection limits around 200 iRBC $\mu L^{-1}$)[5,9]. While these tests are sufficient to diagnose patients with malaria symptoms, they are not sensitive enough to identify asymptomatic carriers of malaria parasites. To identify these will be a stumbling block for regions where the disease is targeted for eradication. Moreover, current MRDTs are unable to quantify parasitemia, which would be beneficial to monitor treatment efficacy and transmission dynamics. Further deficiencies of MRDTs include cross reactions with antigens of other parasites or intrinsic lack of target antigen in the parasite population such as the reported deletion of the hrp2 gene, resulting in false positive or negative results[5,10]. Moreover, the antibodies used in the strips do not always perform as expected at the high and humid temperatures encountered in the regions where malaria is present[8]. As typical MRDTs should be kept below 30 °C[11], an effective cooling chain must be implemented during their transport and storage. Alternatively, highly sensitive methods such as loop-mediated isothermal amplification[12] or enzyme linked immunosorbent assay (ELISA)[13] are also available, and efforts are underway to develop quantitative real-time polymerase chain reaction (qPCRs) for field work[14–16]. However, up to now, none of these techniques can be used as point of care tests. Few malaria diagnostic tests are based on the detection of hemozoin, which is a by-product of hemoglobin digestion by the malaria parasite[17,18]. It is a biocrystal which consists of centrosymmetric μ-propionate dimers of heme (Fig. 1a)[19]. More importantly, hemozoin is a powerful biomarker for malaria infection, since it is found in all erythrocyte stages of all clinically relevant *Plasmodium* species[17,20]. Physical methods such as laser desorption mass spectrometry[21], Raman spectroscopy[22], laser-induced nanobubble formation[20], multiple-angle polarization scatter separation[23], or magnetically induced dichroism[24–26] have been used to detect hemozoin, also in

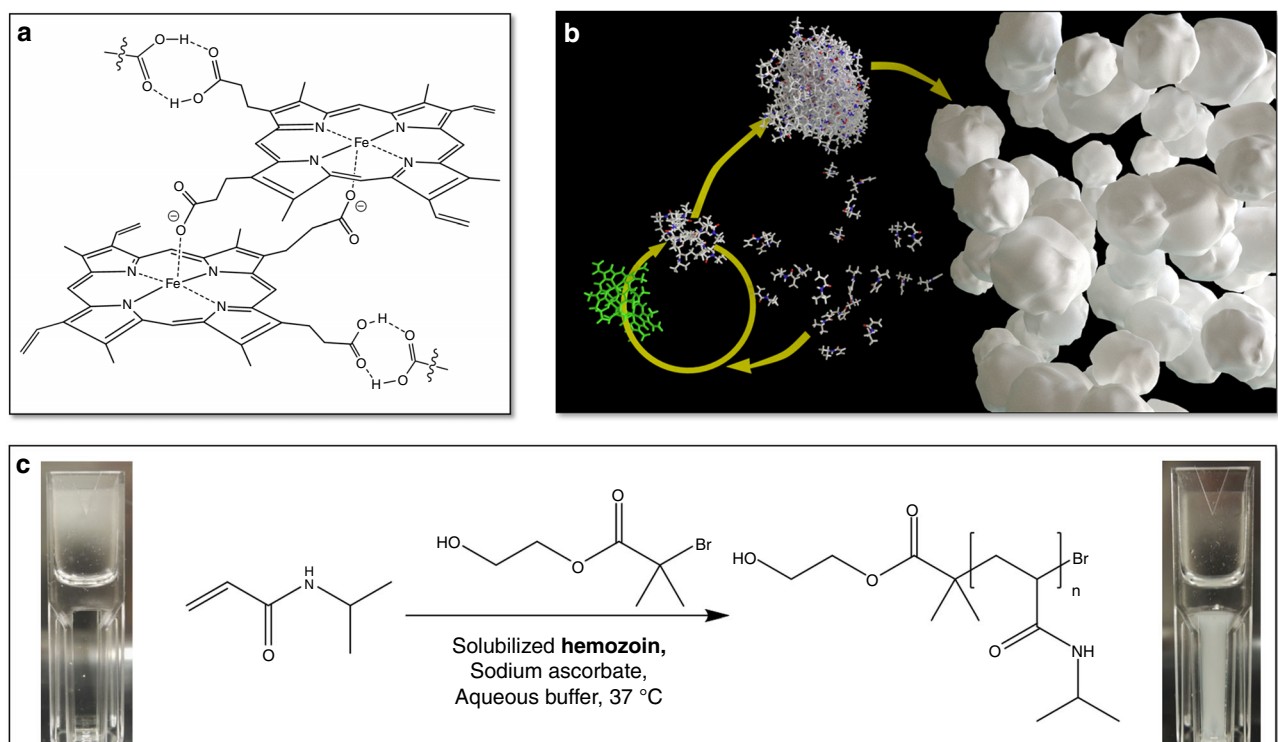

**Fig. 1** Biocatalytic precipitation polymerization of NIPAAm as hemozoin assay. **a** Structure of Hz, a supramolecular crystal of dimers of hematin. **b** Schematic depiction of the Hz-catalyzed precipitation polymerization of NIPAAm. **c** Reaction scheme of the polymerization and photographs of a reaction mixture at the start of the reaction (left) and after 1 h (right) (1000 ng mL$^{-1}$ sHz)

peripheral blood. There are existing efforts to yield these techniques less complex, showing that hemozoin is a valid biomarker for malaria diagnosis.

Radical polymerization reactions are ideally suited to amplify signals in biosensing[27–33] because of the creation of one radical at the site of molecular recognition. This starts a chain reaction that joins many, often several thousands, monomer molecules of small molecular weight into a detectable polymer of high molecular weight. Moreover, if the polymerization is catalyzed by an analyte, one catalyst molecule initiates and mediates the formation of many polymer molecules, achieving even stronger molecular amplification. To our knowledge, solubilized hemozoin has not been explored as a catalyst for radical polymerization, neither for synthesis purposes nor for biosensing applications, even though heme-containing enzymes and hemin[34–36] are known to initiate free radical polymerizations[37–39] or to catalyze reversible-deactivation radical polymerizations[40–47].

Here, we present an ultrasensitive polymerization-based assay for the detection and quantification of hemozoin. Solubilized hemozoin is used to catalyze the radical polymerization of N-isopropylacrylamide (NIPAAm). The formation of poly(N-isopropylacrylamide) (PNIPAAm) can be simply detected by conducting the polymerization at 37 °C, i.e., under conditions in which the polymer precipitates[48,49]. This results in a transparent, colorless liquid becoming turbid (Fig. 1b, c). The formation of turbidity can be visually observed as an indicator for malarial infection and can be quantified by absorbance measurements using simple optical instruments. Biocatalytic precipitation polymerizations were carried out under activator regenerated by electron transfer (ARGET) ATRP conditions, i.e., using an ATRP initiator and an excess of reducing agent. These conditions were not selected to achieve reversible-deactivation radical polymerizations, but to use stable organic reagents that would not initiate the polymerization on their own, even when heated or exposed to sunlight. Moreover, ATRP initiation is a selective reaction that only specific biocatalysts such as hemozoin can achieve, so that interference by other compounds of biological fluids is low. The assay combines the high sensitivity of signal-amplification by catalytic radical polymerizations with cheap and stable non-biological reagents and a simple read-out that does not require sophisticated instrumentation. This renders it ideal for diagnostics in the field, and the assay could become an important tool in the fight against malaria.

## Results

**Precipitation polymerizations catalyzed by solubilized Hz**. Natural hemozoin (nHz), which is insoluble under physiological conditions, was obtained from *Plasmodium falciparum* cultures. It was isolated by magnetic affinity chromatography and solubilized in basic conditions (0.4 M NaOH, pH 13.06). As some of the experiments required larger amounts of hemozoin, synthetic hemozoin (β-hematin; sHz) was also used. sHz is considered a chemical and structural analogue to nHz (Fig. 1a)[19].

Solubilized Hz can catalyze the polymerization of NIPAAm to PNIPAAm in the presence of the ATRP reagents sodium ascorbate (reducing agent) and 2-hydroxyethyl bromoisobutyrate (HEBIB) (initiator) (Fig. 1b). Polymerizations were carried out as precipitation polymerizations above the lower critical solution temperature (LCST) of PNIPAAm[48]. As the reactions proceeded, the polymers precipitated and the transparent solution turned turbid. This process was observable by bare eye (Fig. 1c and Supplementary Video) and was also monitored by extinction measurements in a UV–Vis spectrophotometer. Extinction is the

sum of light scattering and absorption, i.e., –log[T], where T is the transmission at a defined wavelength. After an initial lag phase, the extinction increased in an almost linear fashion over time (Fig. 2a). At longer reaction times the rate of turbidity formation slowed down until the extinction reached a plateau. The rate of extinction increase was fitted with a straight line and the slope (ΔE/Δt) was defined as the quantitative read-out of the assay.

A series of experiments was carried out to optimize the parameters for the assay. Reactions that were carried out in air did not result in a homogeneous turbidity throughout the solution or did not polymerize at all (Supplementary Fig. 1), because oxygen diffused into the reaction mixture and inhibited the polymerization. This problem was solved by sealing the reactions from the atmosphere with a layer of mineral oil. Time-dependent spectral scans of the polymerization reactions revealed that the increase in extinction over time, i.e., the response of the assay, was higher in the UV than in the visible range of the spectrum (Fig. 2a) even though the monomer and the reducing agent absorb below 400 nm (Supplementary Fig. 2a). This is not surprising, as the increase in optical density is caused by the increased scattering of light. However, at high concentrations of hemozoin, the absorption bands of heme, especially the Soret band around 387 nm, also contributed to the spectrum (Supplementary Fig. 2b). In order to minimize this distortion of the assay, extinction measurements were usually carried out at 600 nm (Fig. 2b, c). At very low Hz concentrations the absorption of the hemozoin was negligible so the sensitivity of the assay could be further increased by measuring extinction at 380 nm (Fig. 2d).

If the hemozoin-catalyzed precipitation polymerization were to be used as a quantitative assay to determine hemozoin in a solution, the read-out of the assay should depend on the concentration of hemozoin. Therefore, precipitation polymerizations were carried out with various amounts of solubilized Hz (Fig. 2b and Supplementary Video). A higher concentration of solubilized hemozoin resulted in a faster increase in extinction, as well as a shorter lag phase and a higher turbidity at the end of the reaction. Dose-response curves were obtained by plotting ΔE/Δt against the concentration of solubilized hemozoin (Fig. 2c). For natural hemozoin, the read-out of the assay scaled linearly with the concentration of nHz over a range of ~25–1000 ng mL$^{-1}$ nHz (the highest concentration tested) ($\frac{\Delta E}{\Delta t} = -2.979 \cdot 10^{-5}$s$^{-1}$ $+4.430 \cdot 10^{-6}$s$^{-1} \cdot c_{assay}$(Hz); $R^2 = 0.98$). At lower concentrations of the analyte, the signal scaled with the concentration of nHz, but not linearly. The sensitivity of the assay could be further increased by conducting the extinction measurements at 380 nm. The assay read-out at this wavelength increased linearly with the concentration of nHz at very low amounts of nHz (Fig. 2d). The decision limit and the detection limit of the assay to determine the presence of natural hemozoin with a confidence level of 95% were 0.49 ng mL$^{-1}$ nHz and 0.85 ng mL$^{-1}$ nHz, respectively. Based on calculations given by Newman et al.[50] and Egan et al.[51], we derived an equation to determine the level of parasitemia from the measured concentration of hemozoin (c.f. Supplementary Methods). According to this equation, a concentration of 0.85 ng mL$^{-1}$ Hz at 380 nm would hypothetically correspond to a parasitemia of around 1.4 iRBC μL$^{-1}$ (2.8·10$^{-5}$%) (1 iRBC μL$^{-1}$ corresponds to ~0.6 pg μL$^{-1}$ Hz), showing the potential for the development of a highly sensitive technique for malaria diagnosis based on Hz-catalyzed precipitation polymerization.

The dose-response curve for sHz is similar to the one of nHz (Supplementary Fig. 3), albeit with slightly higher rates of turbidity formation for any given concentration of the catalyst. Natural hemozoin contains unsaturated fatty acids[52]. Most likely, they reduce the catalytic polymerization activity of the heme because they are known to scavenge free radicals[53].

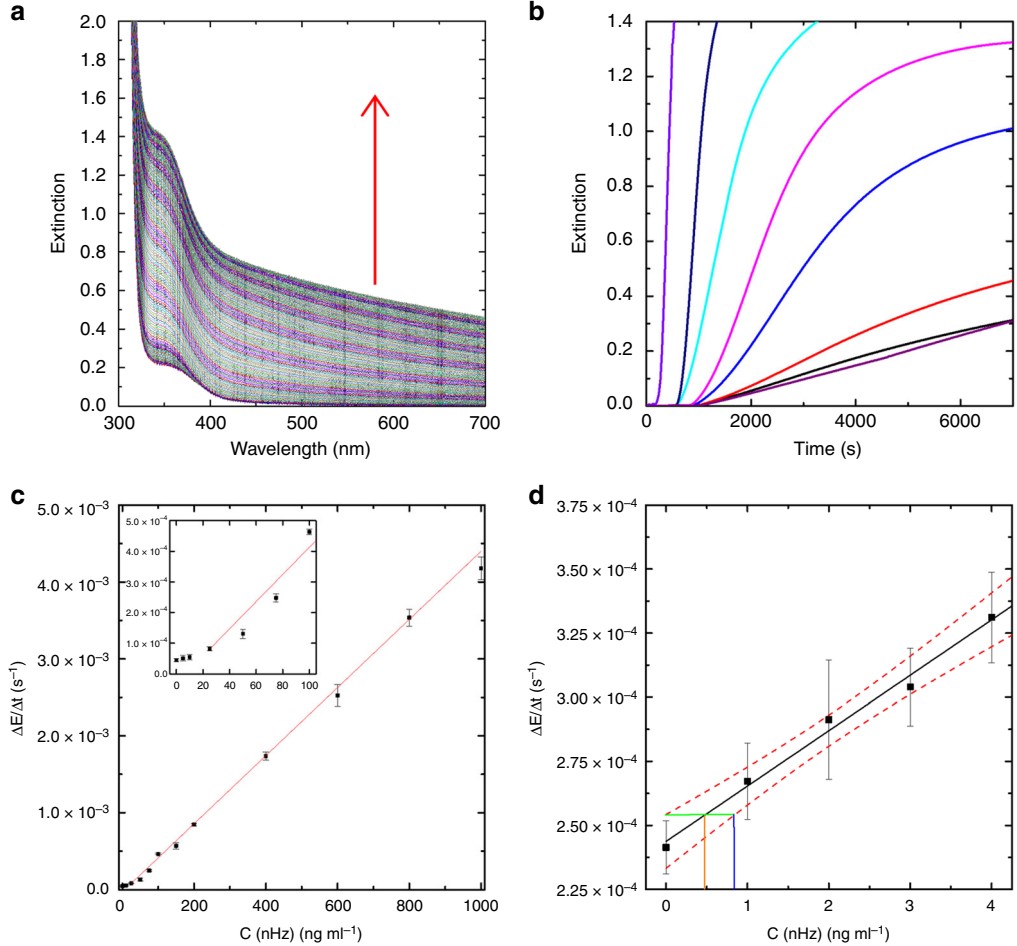

**Fig. 2** Precipitation polymerizations catalyzed by nHz quantified by extinction measurements. **a** Time-dependent UV–vis measurements of a NIPAAm polymerization catalyzed by 25 ng mL$^{-1}$ nHz. The spectra were recorded every 30 s. **b** Time-dependent extinction measurements at 600 nm of NIPAAm polymerization catalyzed by 10 ng mL$^{-1}$ (black), 25 ng mL$^{-1}$ (red), 50 ng mL$^{-1}$ (blue), 100 ng mL$^{-1}$ (pink), 150 ng mL$^{-1}$ (cyan), 400 ng ml$^{-1}$ (navy blue), and 1000 ng ml$^{-1}$ nHz (violet). A blank reaction without nHz is also observed (purple). **c** Dose-response curve for nHz at 600 nm. (Average of $n = 5$ and SD; the red line represents the linear fit to the data between 25 ng mL$^{-1}$ and 1000 ng mL$^{-1}$) **d** Dose-response curve for low concentrations of nHz at 380 nm (Average of $n = 6$ and SD; the black line represents the linear fit to the data, the red dotted lines indicate the 95% confidence interval, the orange vertical line indicates the decision limit and the blue vertical line indicates the detection limit)

The sensitivity of the assay permits the detection and quantification of hemozoin well beyond the limit of UV–vis spectroscopy, a common method to quantify hemozoin in solution[54]. Hz cannot be differentiated by UV–vis spectra from other compounds at concentrations of <100 ng mL$^{-1}$, as none of its characteristic bands can be resolved with certainty (Supplementary Fig. 2). In contrast, the precipitation polymerization assay gives specific and clear read-outs at such low concentrations of Hz.

To demonstrate that the assay can also be conducted in a miniaturized format, hemozoin-catalyzed precipitation polymerizations were carried out in glass capillaries. After a reaction time of 1h, the turbidity in the capillaries was more intense with increasing Hz concentration, while the control reaction without hemozoin remained transparent (Fig. 3). Thus, the assay can also be carried out in just a few microliters of reaction volume, which drastically reduces the required blood sample volume. Moreover, the assay could be further optimized for use in microfluidic lab-on-a-chip devices.

**Detection and quantification of malaria parasites in blood**. To test the potential of the assay as a diagnostic tool, we mimicked patient samples by spiking 1 mL full human blood with varying

quantities of non-synchronized *P. falciparum*-infected iRBCs. As hemoglobin is known to catalyze biocatalytic ATRP reactions[42], it must be removed from the polymerization test. In a first approach, we isolated the biocrystal by selectively lysing red blood cells with saponin which disrupted the RBC membranes while conserving the parasite integrity. This was followed by multiple centrifugation and washing steps that removed water soluble molecules. Then, the parasites were lysed and the cell debris solubilized with SDS, leaving behind the insoluble biocrystals. The obtained samples were tested with our assay. The rate of turbidity formation correlates linearly with the concentration of spiked iRBCs in the blood with a detection limit of 83 iRBCs µL$^{-1}$ and a decision limit of 45 iRBCs µL$^{-1}$ (Fig. 4a). Compared to the theoretical detection limit of the assay based on pure hemozoin (vide supra), this is ~60 times higher. The loss of sensitivity may be due to the depletion of hemozoin throughout the multiple washing steps, or because the calculation of the theoretical detection limit of parasites includes assumptions on the quantity of hemozoin per parasite. Control experiments in which blood samples were not spiked with iRBCs showed comparable turbidity formation rates as the blank reaction. Thus, the sample preparation quantitively removes all interfering biomolecules,

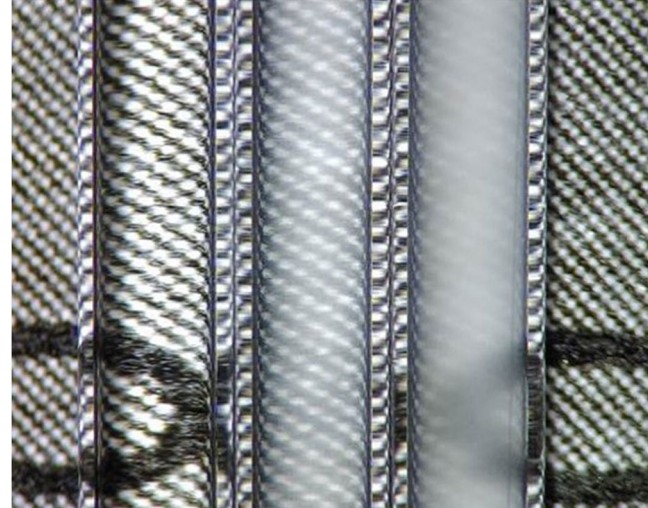

**Fig. 3** Hz-catalyzed precipitation polymerization of NIPAAm in glass capillaries. Microscopic photography (×20 magnification) of polymerizations catalyzed by 0 ng mL$^{-1}$ (left), 400 ng mL$^{-1}$ (center) and 1000 ng mL$^{-1}$ (right) sHz after a reaction time of 1 h. The background is a printed paper

especially hemoglobin, which could catalyze the polymerizations, leaving hemozoin as the only polymerization active compound isolated from blood.

While the previous isolation method extracts hemozoin from blood, a purification method that isolates intact malaria parasites from blood should enhance the sensitivity of the assay, as non-digested hemoglobin in the parasites together with non-crystallized heme would also catalyze the polymerizations. Moreover, this would prevent false positive results caused by some hemozoin lingering in the blood of a patient after an infection has been cured[55]. To test this hypothesis, blood was spiked with iRBCs, and the erythrocytes of 50 µL samples were selectively lysed with saponin. The lysis was followed by size exclusion chromatography using Sepharose as the stationary phase in hand-driven chromatography cartridges. The isolates were centrifuged and the resulting pellets, composed of membrane debris together with parasites, were subjected to the polymerization tests (Fig. 4b). The read-out of the assay scaled with the concentration of iRBCs in blood. The detection limit is 9.7 iRBCs µL$^{-1}$ with a decision limit of 5.3 iRBCs µL$^{-1}$, i.e., a greatly improved sensitivity compared to the sample preparation via isolation of hemozoin. It should be noted that control reactions without iRBCs showed lower turbidity rate formations in the model reactions catalyzed by pure hemozoin. This could be due to the presence of cell membranes that might quench radicals. In conclusion, the isolation of parasites from blood allows the precipitation polymerization assay to detect very low levels of parasitemia in parasite-spiked human blood.

**Fundamentals of Hz-catalyzed precipitation polymerizations**. All the hemozoin-catalyzed precipitation polymerizations followed the same sequence of events. After a lag phase, the turbidity increased until it reached a plateau. To gain fundamental understanding of these three phases, a series of experiments was conducted (Fig. 5). Turbidity is the direct consequence of suspended particles that scatter light. To characterize the particles that formed during the reactions, hemozoin-catalyzed polymerizations were followed by dynamic light scattering (DLS). The average diameter of the particles increased linearly to 1100 nm during the reaction, meaning that more polymer chains

assembled to the particles throughout the experiment (Fig. 5a). Different concentrations of solubilized hemozoin resulted in similar particle sizes (Supplementary Fig. 4a). As the rate of turbidity formation in extinction measurements increased with the concentration of solubilized hemozoin (Supplementary Fig. 4b), it can be concluded that higher concentrations of solu-bilized hemozoin yielded higher numbers of similarly sized polymer particles.

Although the assay was covered with a layer of mineral oil to exclude atmospheric oxygen during the reaction, the solutions were not de-oxygenated to allow for an easy handling of the reagents. Oxygen in the solution could therefore be responsible for the lag phase. To test this hypothesis, solutions were purged with argon prior to the reaction. Turbidity was observed immediately after the reagents had been mixed (Fig. 5b). Moreover, $\Delta E/\Delta t$ was slightly higher than in standard assays. These findings confirm that oxygen in the aqueous solutions was the cause of the lag phase. It is possible that oxygen may have inhibited radical polymerization until it was consumed by the newly-formed radicals in solution. At the same time, this process decreased the concentration of the initiator and therefore caused a slower rate of turbidity formation. While oxygen-free condi-tions would allow for a more sensitive hemozoin assay, improvements in reaction rate and shorter assay times are counterbalanced by a more complex experimental set-up when working under stringent anaerobic conditions. The assay is intended to be used in the field. As a consequence, we decided to rely on the in situ consumption of oxygen by the reaction.

The extinction measurements show that the plateau in turbidity was reached when the reaction stopped. $^1$H NMR experiments were performed to measure the conversion in this phase of the reaction. The monomer to polymer conversion was <5%, i.e., below the detection limit of NMR spectroscopy, for reactions catalyzed by 1000 ng mL$^{-1}$ sHz after a reaction time of 2000 s (Supplementary Fig. 5). To elucidate why the reaction did not proceed to higher conversions, aliquots of fresh reagents were added to reactions after they had reached a constant turbidity (Fig. 5c). Polymerizations could be restarted upon addition of sHz but did not proceed further when new monomer or reducing agent were added. Thus, it appears that the catalyst loses its activity, which may be due to the co-polymerization of the vinyl groups of the catalyst (two per heme unit) with NIPAAm and subsequent precipitation of the copolymer chains. To prove this hypothesis, NIPAAm polymers catalyzed by sHz were synthe-sized. UV–Vis spectra of purified polymers show an absorption band at 387 nm, the Soret band of solubilized hemozoin (Supplementary Fig. 6). The presence of heme in purified polymers was further proven by gel permeation chromatography (GPC), which showed the polymer peak both in the refractive index detector and the UV–vis detector (Supplementary Fig. 7). This is consistent with a previous report that heme can be copolymerized with vinyl monomers[56]. Thus, the reaction resulted in the chemical incorporation of the heme into the precipitating polymer which removed the catalyst from solution. Nevertheless, only small quantities of polymers are required for the formation of turbidity, meaning the assay is highly sensitive to hemozoin.

Although the reaction was carried out under conditions of ARGET ATRP, i.e., with an ATRP initiator and a surplus of reducing agent, we did not expect them to be reversible-deactivation radical polymerizations, because the reactions were carried out in a heterogeneous system in which the polymer chains precipitated throughout the polymerization process. GPC revealed that a polymer that was synthesized with 1000 ng mL$^{-1}$ sHz and purified from reactions after 60 min had a number average molecular weight of 47000 g mol$^{-1}$ and a dispersity of

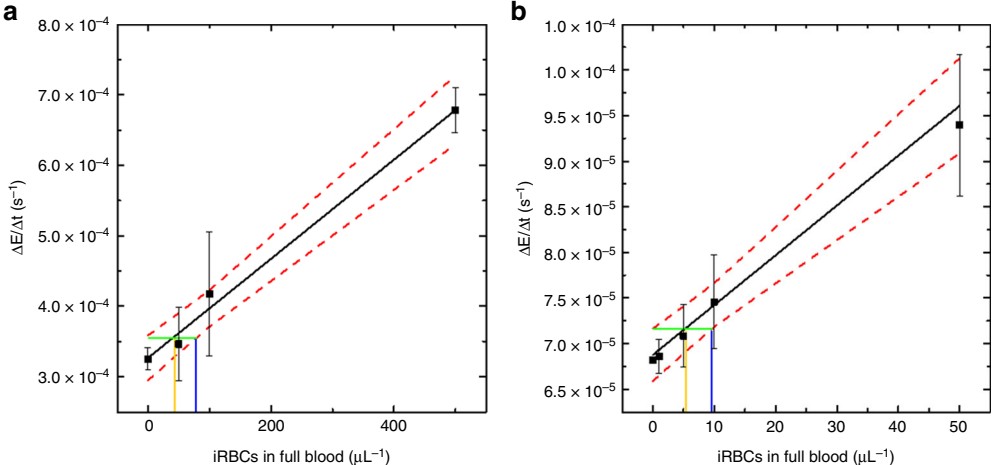

**Fig. 4** Detection and quantification of *P. falciparum* in full human blood. **a** Dose-response curve for precipitation polymerizations of NIPAAm catalyzed by hemozoin that was isolated from blood by multiple centrifugation and washing steps (Average of $n = 5$ and SD). **b** Dose-response curve for precipitation polymerizations of NIPAAm catalyzed by the content of intact malaria parasites (i.e., hemozoin, hemoglobin and free heme) that were extracted from full blood by selective lysis of red blood cells, followed by size exclusion chromatography (Average of $n = 3$ and SD) (the black lines represent the linear fit to the data, the red dotted lines indicate the 95% confidence interval, the orange vertical lines indicate the decision limits and the blue vertical lines indicate the detection limits)

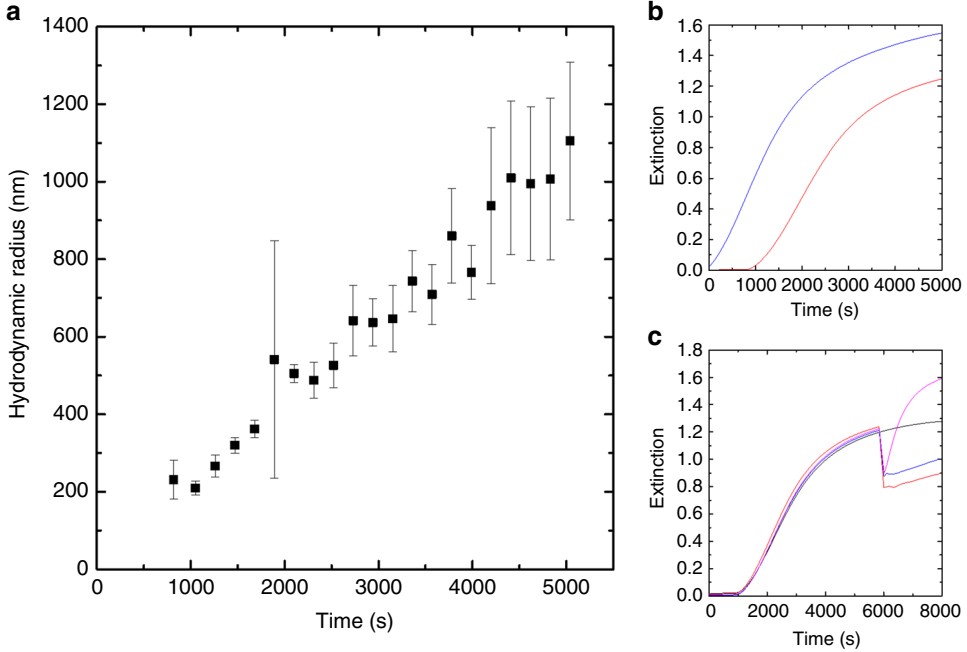

**Fig. 5** Characterization of Hz-catalyzed precipitation polymerizations of NIPAAm. **a** Dynamic light scattering analysis of particle size for a polymerization catalyzed by 25 ng mL$^{-1}$ sHZ (mean values determined as described in the experimental section, SD). **b** Comparison of polymerizations catalyzed by 100 ng mL$^{-1}$ sHz purged with argon (blue) and conducted under normal conditions (red). **c** Effect of the addition of reagents during the plateau phase of the polymerization catalyzed by 100 ng mL$^{-1}$ sHz (at 5500 s): typical experiment (black), addition of initiator (50 µL of a 0.375 M HEBIB solution) (red), addition of monomer and reducing agent (200 µL of a 1.096 M NIPAAm and 0.110 mM NaAsc solution) (blue) and addition of biocatalyst (49.3 µL of a 2.03 µg mL$^{-1}$ sHz solution) (pink). The decrease in extinction after the addition of the reagents is the consequence of the dilution of the reaction mixture

3.8. These are typical values for a polymer obtained by free radical polymerization (Supplementary Fig. 7). The only source of radicals in the reaction mixture was the ATRP initiator, which suggests that the reaction was initiated by an atom transfer of the bromine from the initiator to the catalyst.

**Temperature stability of assay reagents.** If this assay is going to be used in the field, it has to be stable in the hot conditions during transport and storage in tropical or subtropical countries. Therefore, we investigated the stability of all reagents in ageing experiments over a period of 2 months at 50 °C (Supplementary Figs. 8, 9 and 10). $^1$H-NMR spectra before and after this extended exposure to elevated temperatures did not reveal any differences, i.e., the compounds did not appear to degrade and were stable. This hints towards a long shelf-life of the assay even at high temperatures, which represents an important advantage over current MRDTs or primers for PCR.

## Discussion

Current malaria diagnostics are based on microscopy or PCR, which are expensive and labor intensive, or depend on MRDTs, which lack sensitivity for low levels of parasitemia. Thus, alternative assays are of high interest. The approach herein uses the thermo-responsive properties of PNIPAAm in combination with double signal amplification by catalytic radical polymerization to detect the presence of Hz (solubilized in the assay to heme). Upon Hz-catalyzed precipitation polymerization of NIPAAm above the LCST of the polymer, the reaction mixture becomes turbid. This can be seen qualitatively by unaided eye or recorded quantitatively by turbidity measurements. The rate of turbidity formation scales linearly with the concentration of solubilized hemozoin, which makes the assay quantitative. The assay can detect as low as 0.85 ng mL$^{-1}$ Hz in model conditions and could differentiate 10 iRBC µL$^{-1}$ in parasite-spiked full blood. Though the assay remains to be studied with various parasite stages, in field conditions and with the blood of a variety of patients, the results indicate that its sensitivity could be up to 20 times higher than current MRDTs. The assay is suited for resource-limited settings as it relies on a simple read-out and inexpensive chemicals. The reagents are also temperature stable, avoiding the necessity for cool chain storage in the hot environments where malaria is endemic. However, before the assay can be employed and tested as a point of care test, further development is required, especially to engineer a simple-to-use device to extract hemozoin or parasites from blood samples. Hemozoin is present throughout the erythrocyte life cycle of all malaria parasite species including the sexual gametocyte forms[17,20]. Thus, it is a pan-species malaria biomarker. Though identification of malaria parasite species is important for the administration of appropriate treatment schemes, the assay could be employed to screen populations for asymptomatic carriers with low levels of parasitemia. The parasite species would then only have to be confirmed by other methods for those cases that have been tested positive with the assay. Therefore, the assay could become a powerful tool in the fight against malaria. The reactions reported herein represent a biosensing assay based on the precipitation polymerization of a polymer coupled to turbidity measurements. This sensing principle can be applied to detect and quantify other analytes, such as oxidoreductases and peroxidases, through their highly selective reaction with alkyl halide initiators. Further efforts will be carried toward the validation of the biocatalytic hemozoin sensing assay in clinical settings and toward the implementation of the assay into a portable instrument format.

## Methods

**Biocatalytic precipitation polymerization assay.** Unless otherwise stated, in a typical experiment 1.429 g (12.63 mmol) of NIPAAm were weighted together with 250 mg (1.26 mmol) of (+)-sodium L-ascorbate into a round-bottom flask. Thereafter, 10 mL of sodium phosphate buffer solution (pH 6.5, 0.12 M) were added to obtain a stock solution which was employed for the polymerization reactions. 0.700 mL of this stock solution were pipetted into disposable semi-micro poly(methyl methacrylate) cuvettes (path length: 1 cm). Various volumes of solubilized hemozoin solutions in 0.4 M NaOH were added and adjusted to a total volume of 50 µL of 0.4 M NaOH. The volume of buffer was adjusted to achieve a total volume of 950 µL in each cuvette with a pH of 6.9. The solutions were then sealed against ambient oxygen by adding 500 µL of mineral oil. The cuvettes were incubated for 3 min at 37 °C. Finally, the polymerization was initiated by the addition of 50 µL of a 375 mM 2-hydroxyethyl HEBIB solution in DMF. The final concentration of reagents in the assay was: 767 mM NIPAAm, 18.7 mM HEBIB and 76.7 mM sodium ascorbate, i.e., a ratio of 41:1:4.1. The cuvette was placed in a thermostatted cuvette holder (37 °C) of a UV–vis spectrometer and extinction was recorded at 600 nm or at 380 nm every 30 s with an integration time of 1 s for up to 4 h.

## Data availability

The datasets generated during and analyzed during the current study are available in the Zenodo repository, DOI 10.5281/zenodo.2541603.

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

## Acknowledgements

This work was supported by the Swiss National Science Foundation through projects PP00P2_144697 and PP00P2_172927, through the National Centre of Competence in Research (NCCR) Bio-Inspired Materials, and a BRIDGE Proof-of-Concept grant (20B1-1_173771). Moreover, financial support by the Novartis Foundation for Biological-Medical Research, and the Gebert Rüf Foundation is gratefully acknowledged. We thank Katherine A. de Villiers (Stellenbosch University) for fruitful discussions, Samuel Lörcher (University of Basel) for the GPC analysis, Dominic Urban (Adolphe Merkle Institute) for ICP-OES measurements, Miguel Spuch-Calvar (Adolphe Merkle Institute) for the artwork, and Alexey Anatolyevich Larionov together with Patricia Matthey (Department of Medicine, University of Fribourg) for phlebotomy.

## Author contributions

O.R.-G., J.P., and S. Ra contributed equally to the work. O.R.-G., J.P., and N.B. conceived the project. O.R.-G., J.P., S. Ra, and N.B. designed and interpreted most of the experiments. O.R.-G., J.P., and S.Ra performed most of the experiments. S.B. performed and interpreted the light scattering experiments. S.Ru and M.A.H.-C. provided cultured parasites, natural hemozoin, and other biological samples. P.-Y.M. and H.-P.B. provided expertise on malaria and supervised S. Ru and M.A.H.-C. N.B. supervised the project. O.R.-G., J.P., S.Ra and N.B. wrote the manuscript. All authors edited the manuscript.

## Additional information

**Competing interests:** O.R.-G., J.P., and N.B. declare that they have filed one patent on the technology described herein, and that they are in the process of founding a spin-off company to bring the assay into the market. The remaining authors declare no competing interests.

