## [Peer Review File · Nature Communications]

Reviewers' comments:

Reviewer #1 (Remarks to the Author):

Rifaie-Graham et al. have reported a new method to detect hemozoin in aqueous solutions. This method involves the use of hemozoin as a catalyst in a radical-mediated polymerization reaction. The polymer that forms precipitates, leading to a turbid solution. The central claim of the paper and reasoning for why it should be published in Nature Communications is as follows:

"The rate of turbidity increase is proportional to the concentration of hemozoin, with a detection limit of 0.85 ng mL (1.4 parasites / μ L of blood). The assay is 140 times more sensitive than other malaria rapid diagnostic tests and could be applied as a point of care test in the field."

In order for this claim and the comparison with other methods to be valid and convincing, the authors must conduct their analysis starting from a relevant bodily fluid (blood, serum or plasma) as research teams who have reported the competing methods have all done. The available data do not support the claims. Many methods are exquisitely sensitive at detecting the analyte in simple aqueous solutions, and yet when the sample matrix becomes more complex and realistic, the detection limit goes up as does the incidence of false positives.

Reviewer #2 (Remarks to the Author):

This is an excellent piece of work on the utilization of the LCST-type behavior of PNIPAAm formed by hemozoin mediated ATRP. It is strongly recommended to be published after some minor revision on the basis of comments below.

COMMENTS

1. Considering the extinction-time plots in Figure 2b, Figures S1-S4 and the absorption spectra in Figures S6 and S11, in reality the 0.85 ng/mL detection limit would not be realistic. The extinction curve in Figure 2b overlaps with the 10 ng/mL curve, and the components have absorption at the applied wavelenghts. The authors are proposed to discuss this issue.
2. The "controlled/living" term should be deleted. Even those who proposed this term use it very seldom or do not use it at all anymore.
3. It is an obvious question whether this process is selective enough under practical conditions. As known, hemoglobin in blood catalyze the polymerization of NIPAAm (ref. 38 in this manuscript). This would mean that the polymerization in a normal blood sample would not be selective for hemozoin. What is the opinion of the authors on this matter?
4. The correct bibliographical data for Ref. 45: Macromol. Chem. Phys. 218, 1600470 (2017)

Reviewer #3 (Remarks to the Author):

This manuscript describes the use of base solubilised haemozoin as a redox catalyst for initiation of a polymerisation reaction producing an insoluble polymer that can be used to qualitatively and quantitatively determine the presence of ferrihaem released as a consequence of haemozoin

solubilisation. The authors propose this process as a new malarial diagnostic.

In principle this is a novel approach that could generate much interest. It is an innovative and thought-provoking method for the detection of haemozoin and is much more sensitive than existing physical techniques for haemozoin detection. As such, it could potentially be suitable for publication in this journal. On the other hand, several substantial hurdles still need to be overcome before I would consider this work suitable for publication.

1. A fundamental disadvantage of using haemozoin as a diagnostic target is that it is not species-specific with respect to the different *Plasmodium* species. Of course, this can also be an advantage, but needs some additional mention of advantages and disadvantages.

2. A more serious disadvantage of haemozoin as a diagnostic tool is that it lingers in the host long after malarial infection has been cured. This is a result of the release of haemozoin into the host every time a red cell ruptures upon completion of a blood cycle. This also needs careful discussion and justification.

3. A much more serious problem will require further experimental work in my opinion. The authors have done a considerable amount of pre-processing of parasite cultures before measuring haemozoin levels using their assay. This includes magnetic concentration of parasitised red blood cells, centrifugation steps, selective saponin lysis of RBCs, freeze-thaw lysis of trophozoites and collection and washing of haemozoin. In a real matrix of human blood, the process would be similar, but likely might require additional steps to prevent clotting and collect RBCs. An approach that would directly allow tests on blood needs further development so that these steps can be conducted in the field without sophisticated lab equipment.

4. Furthermore, in the case of *P. falciparum*, trophozoites and schizonts are sequestered in capillaries, so levels of RBC haemozoin in the circulation would be associated with only the ring stage and would be very low, probably resulting in substantially higher detection limits. Much of the haemozoin in fact is present in white blood cells (monocytes/granulocytes) and would require other methods of isolation.

For a genuine diagnostic assay, these issues need to be fully addressed. In other words, steps such as centrifugation would need to be avoided and lysis and washing simplified. Absent of these, the assay represents an interesting step towards a diagnostic method, but no more than that. If the authors can extend the work to a state where the method can be directly applied to blood samples it would be an extremely valuable contribution and publication in this journal would be fully justified.

Please find a point-by-point response to the reviewer comments below. All important changes in the manuscript are highlighted in yellow in the attached files.

Reviewers' comments:

Reviewer #1 (Remarks to the Author):

Rifaie-Graham et al. have reported a new method to detect hemozoin in aqueous solutions. This method involves the use of hemozoin as a catalyst in a radical-mediated polymerization reaction. The polymer that forms precipitates, leading to a turbid solution. The central claim of the paper and reasoning for why it should be published in Nature Communications is as follows:

"The rate of turbidity increase is proportional to the concentration of hemozoin, with a detection limit of 0.85 ng mL (1.4 parasites / μ L of blood). The assay is 140 times more sensitive than other malaria rapid diagnostic tests and could be applied as a point of care test in the field."

In order for this claim and the comparison with other methods to be valid and convincing, the authors must conduct their analysis starting from a relevant bodily fluid (blood, serum or plasma) as research teams who have reported the competing methods have all done. The available data do not support the claims. Many methods are exquisitely sensitive at detecting the analyte in simple aqueous solutions, and yet when the sample matrix becomes more complex and realistic, the detection limit goes up as does the incidence of false positives.

We thank Reviewer 1 for this very accurate comment. Indeed, many other groups go a step beyond the detection of the analyte in model conditions and test the assay in relevant body fluids. We therefore developed two quantitative protocols for the isolation of the biomarker from *P. falciparum* spiked blood samples. The spiked blood samples mimic infected blood, which is difficult to obtain from real patients in non-endemic countries such as Switzerland.

The first protocol isolates the biocrystal hemozoin which is insoluble in water. This solubility difference can be used to isolate the crystals by a combination of lysis and washing steps. The method also employs surfactants which do not dissolve the crystals. The detection limit is at 83 parasites μ L⁻¹ which is higher than expected from experiments with pure hemozoin, but still significantly lower than the limit of detection of malaria rapid diagnostic tests (MRDTs). Importantly, the experiments show that ATRP-active molecules that are not associated with parasites, such as the hemoglobin of non-infected red blood cells, can be completely removed from the samples and therefore do not interfere with the assay.

The second protocol aims to isolate the full parasite from surrounding hemoglobin. The assay benefits from this approach as it permits to isolate not only the hemozoin but also other compounds in the parasites such as non-digested hemoglobin and non-crystallized heme which also catalyze the polymerizations together with hemozoin. The data shows that 9.7 parasites μ L⁻¹ of blood could be isolated employing this approach, which is 20-fold lower than the detection limit of MRDTs. Thus, our hemozoin assay is in the clinical relevant range for the detection of non-symptomatic carriers of malaria parasites.

Reviewer #2 (Remarks to the Author):

This is an excellent piece of work on the utilization of the LCST-type behavior of PNIPAAm formed by hemozoin mediated ATRP. It is strongly recommended to be published after some minor revision on the basis of comments below.

COMMENTS

1. Considering the extinction-time plots in Figure 2b, Figures S1-S4 and the absorption spectra in Figures S6 and S11, in reality the 0.85 ng/mL detection limit would not be realistic. The extinction curve in Figure 2b overlaps with the 10 ng/mL curve, and the components have absorption at the applied wavelenghts. The authors are proposed to discuss this issue.

This is an accurate review by Reviewer 2. It is true that in the figures depicted by the Reviewer, the reactions catalyzed by solubilized hemozoin at concentrations near 5 ng mL⁻¹ start to become similar to the blank reactions in the absence of catalyst. However, the reactions in these figures were recorded at a wavelength of 600 nm. The reason for this was that many biomolecules including solubilized hemozoin absorb in the UV and near-to-UV wavelength spectrum. Thus, to observe the evolution of the polymerization reactions without interferences on the transformation of biomolecules we chose higher wavelengths where such compounds do not absorb light.

As light scattering is enhanced at lower wavelengths and the formation of turbidity relies on the formation of PNIPAAm nano- and microparticles, we sought to increase the detection limit by measuring the extinction rates at 380 nm. Thus, the detection limit of 0.85 ng mL⁻¹ was stated for reactions carried at such wavelengths. This has been clarified in the text.

2. The "controlled/living" term should be deleted. Even those who proposed this term use it very seldom or do not use it at all anymore.

We thank the reviewer for raising this point. This has been corrected in the text.

3. It is an obvious question whether this process is selective enough under practical conditions. As known, hemoglobin in blood catalyze the polymerization of NIPAAm (ref. 38 in this manuscript). This would mean that the polymerization in a normal blood sample would not be selective for hemozoin. What is the opinion of the authors on this matter?

We agree with the comment of Reviewer 2 which seems to be a similar concern for Reviewer 1 and 3. This was indeed one of the main drawbacks of this work in its first version. Thus, two methods for the isolation of the biocrystal hemozoin from full blood samples were developed for this revision showing that hemozoin can be quantitatively isolated from whole blood and while not extracting any other unwanted ATRP-active compounds from the samples. Thus, we are confident that the hemozoin assay, in combination with a sample work-up to isolate hemozoin or parasites will enable the realization of a real-world diagnostic tool. The text was changed.

4. The correct bibliographical data for Ref. 45: Macromol. Chem. Phys. 218, 1600470 (2017).

Many thanks for this comment. This has been corrected.

Reviewer #3 (Remarks to the Author):

This manuscript describes the use of base solubilised haemozoin as a redox catalyst for initiation of a polymerisation reaction producing an insoluble polymer that can be used to qualitatively and quantitatively determine the presence of ferrihaem released as a consequence of haemozoin solubilisation. The authors propose this process as a new malarial diagnostic.

In principle this is a novel approach that could generate much interest. It is an innovative and thought-provoking method for the detection of haemozoin and is much more sensitive than existing physical techniques for haemozoin detection. As such, it could potentially be suitable for publication in this journal. On the other hand, several substantial hurdles still need to be overcome before I would consider this work suitable for publication.

1. A fundamental disadvantage of using haemozoin as a diagnostic target is that it is not species-specific with respect to the different Plasmodium species. Of course, this can also be an advantage, but needs some additional mention of advantages and disadvantages.

We thank Reviewer 3 for this thorough comment. We agree that the lack of species differentiation is a disadvantage of our test, as treatment schemes are species-dependent. However, the test allows detecting very low parasitemia levels with cheap chemicals. We envision that this test could be used as a mass screening for asymptomatic reservoirs with low levels of parasitemia. If positives to this test would be found, then PCR or other methods could be employed to confirm the species. We have clarified this in the text.

2. A more serious disadvantage of haemozoin as a diagnostic tool is that it lingers in the host long after malarial infection has been cured. This is a result of the release of haemozoin into the host every time a red cell ruptures upon completion of a blood cycle. This also needs careful discussion and justification.

This is a good thought and has already been highlighted in reviews such as Delahunt et al.: doi:10.1186/1475-2875-13-147. However, this disadvantage is highly dependent of the hemozoin isolation method from blood. Methods that would allow the selective isolation of the malaria parasites from the rest of the components of the blood would overcome this disadvantage, as the presence of hemozoin (together with non-digested hemoglobin and non-crystallized heme) would be directly related to an integral parasitic structure. One of the methods that we have developed relies on the isolation of the parasite from the rest of blood components and therefore fulfils this criterion.

3. A much more serious problem will require further experimental work in my opinion. The authors have done a considerable amount of pre-processing of parasite cultures before measuring haemozoin levels using their assay. This includes magnetic concentration of parasitised red blood cells, centrifugation steps, selective saponin lysis of RBCs, freeze-thaw lysis of trophozoites and collection and washing of haemozoin. In a real matrix of human blood, the process would be similar, but likely might require additional steps to prevent clotting and collect RBCs. An approach that would directly allow tests on blood needs further development so that these steps can be conducted in the field without sophisticated lab equipment.

We agree to the comment and we have elaborated on this direction. We have proposed two methods of isolation ATRP-active biomarkers from parasite-spiked full blood samples which are now

implemented in the main manuscript and the supplementary information. The first method relies on the solubility differences of hemozoin with the rest of blood components. The isolation was performed by multiple centrifugation and washing steps. The results prove that hemozoin can be quantitatively isolated from blood and introduced into the polymerization tests. Though the isolation is time consuming, it can be performed with one piece of equipment that can be found in low resource settings. The second approach relies on a one step lysis and isolation of the parasites by size exclusion column chromatography, using hand-operated chromatography cartridges. The isolates were then concentrated by a centrifugation step prior to their injection in the test. The results show that relative simple steps can lead to a quantitative assay result, starting from 50 μ l of full blood. We are confident that further technological development, such as the integration of the sample purification in a lateral flow or microfluidic cartridge format, is feasible. However, this is more an engineering problem, while the manuscript describes the scientific foundation for a novel biosensing method.

4. Furthermore, in the case of *P. falciparum*, trophozoites and schizonts are sequestered in capillaries, so levels of RBC haemozoin in the circulation would be associated with only the ring stage and would be very low, probably resulting in substantially higher detection limits. Much of the haemozoin in fact is present in white blood cells (monocytes/granulocytes) and would require other methods of isolation.

Even though the presence of microscopic hemozoin crystals is typical for trophozoites and later stages, the biocrystal will be present on the nanoscale in ring stages as it starts to form upon hemoglobin digestion. Moreover, there is a current thought that the malaria stage that should be able to be diagnosed is the gametocyte. This is due to gametocytes being the only stages that in the case of being phlebotomized by mosquito vectors, will be able to evolve to the sporozoite infecting stages. Gametocytes contain larger amounts of hemozoin than trophozoite late stages. Thus, the test should be sensitive enough to detect low levels of gametocytes.

The size exclusion chromatography method that we have proposed was performed with non-synchronized cultures spiked into blood samples. Such samples contained a large fraction of ring stages. The detection of such stages was allowed by the isolation of the full parasite which contained non-digested hemoglobin, non-crystallized heme and low amounts of hemozoin. As a result, high sensitivities were achieved.

We agree that the performance of the assay will depend greatly on the stages of the parasites and that an in depth correlation of hemozoin/parasite load in peripheral blood to parasite stage and parasite species will be needed to fully assess the performance of the assay. However, this will be a substantial study that goes well beyond the scope of this manuscript, which focuses on introducing a new kind of sensing principle to malaria diagnostics. We have carefully worded the conclusions to address these uncertainties.

For a genuine diagnostic assay, these issues need to be fully addressed. In other words, steps such as centrifugation would need to be avoided and lysis and washing simplified. Absent of these, the assay represents an interesting step towards a diagnostic method, but no more than that. If the authors can extend the work to a state where the method can be directly applied to blood samples it would be an extremely valuable contribution and publication in this journal would be fully justified.

We thank the reviewer for this encouraging comment. We are convinced that, with the development of relative simple work-up protocols to extract hemozoin and parasites from full blood, we have achieved a major improvement in the report and demonstrate that the precipitation polymerization-based assay is capable to detect malaria parasites in full blood. We are currently working towards implementing sample work-up and turbidity assay in a lateral flow/microfluidic lab-on-a-chip format. This will be reported in future publications.

Reviewers' comments:

Reviewer #1 (Remarks to the Author):

The new experiments that the authors have conducted with whole blood spiked with cultured *P. falciparum* strengthen the work considerably.

I recommend the following revisions (no further experiments) prior to publication:

1. At this stage of development, it is more appropriate to compare this new method with other laboratory-based tests than with rapid detection test (RDTs, e.g. lateral flow).

I have great appreciation for the advances the authors have made since the initial submission. However, a great deal of work (including conceptual advances to get around trade-offs-- performance vs. cost, time, equipment, skilled labor, etc.) still remain before this test will be comparable in simplicity to RDTs.

The hand-driven chromatography approach quoted below still involves many steps, a lot of time, and substantial laboratory equipment and technical expertise:

"Parasite isolation by size exclusion chromatography: 10 mL of human blood were collected in heparin-coated tubes (Sarstedt Monovette Li-Heparin LH/9 mL). 50 μ L of this sample were transferred into a 1.5 mL Eppendorf tube (polypropylene) and spiked with known quantities of a nonsynchronized *P. falciparum* culture (concentration of parasites determined by Giemsa staining). The RBC membranes were selectively lysed with 250 μ L of a 0.3 % (w/v) solution of saponin in PBS. The lysate was submitted to a size exclusion column consisting of Sepharose 4B CL packed in a Biotage SNAP Ultra 10 g column as the stationary phase with PBS as the mobile phase. Prior to the chromatography, the columns were equilibrated by substituting the 20:80 ethanol/water solution in which the stationary phase is delivered with 50 mL of ultrapure water and then 50 mL of PBS buffer. Then, the lysate was loaded to the column and eluted by passing 6 mL PBS through the column by means of a hand-driven syringe., The first 5 mL of PBS were collected and contained the parasites. A clear band for the presence of hemoglobin was observed and remained on the column, which was afterwards discarded. The isolate was centrifuged in a 15 mL Falcon tube at 5000 x g for 10 min forming a pellet. The supernatant was discarded and the pellet was dried overnight in a vacuum oven at 60 °C. The parasites were further lysed and their contents solubilized in 55 μ L of a 0.4 M NaOH aqueous solution and sonicated for 30 min (sonicator bath Sonoswiss SW3). The sample was then centrifuged for 10 min at 5000 x g to spin down the sample and 50 μ L were introduced into the polymerization assay."

I agree with the authors' enthusiasm for making their new test competitive with RDTs, but that future work is non-trivial. At present, it is more appropriate to compare with laboratory-based tests. While I don't think further experimental work is required prior to publication, the writing/framing of the current accomplishment should be revised prior to publication.

2. Parasites μ L⁻¹ and infected red blood cells μ L⁻¹ are both used in various places in the manuscript. A clear statement on how to relate these quantities to one another and to ng/mL hemozoin would be helpful in the manuscript (the earlier, the better).

3. These competing methods:

"Physical methods such as laser desorption mass spectrometry,²¹ Raman spectroscopy,²² laser-induced nanobubble formation,²⁰ multiple-angle polarization scatter separation,²³ or magnetically induced dichroism²⁴⁻²⁶ have been used to detect hemozoin, also in peripheral blood. But none of these methods are used widely due to their technical complexity and/or lack of sensitivity."

could be discussed in a bit more depth, including efforts to make user-friendly, field portable sensors/devices.

It is not necessary to cite these particular studies, but they are offered as examples of the kinds of efforts that should not be dismissed without thoughtful consideration:

<https://pubs.acs.org/doi/10.1021/acssensors.8b00269>

<https://www.nature.com/articles/nm.3622>

Reviewer #2 (Remarks to the Author):

The authors have carried out detailed revision of their manuscript on the basis of the reviewers' comments. It is recommended to be accepted for publication after some minor revision on the basis of the comment below.

As a result of Comment 2 by Reviewer 2, the authors use the term of "reversible-deactivation radical polymerization" instead of the sloppy undefined "controlled/living" term for polymerizations catalyzed by heme containing enzymes or hemin in their revised manuscript. The term "reversible-deactivation polymerization" is even worse than "controlled/living polymerization". As described in every polymer chemistry textbook, chain polymerization, including radical polymerizations as well, proceeds by propagation, and never by "reversible-deactivation". When there is an equilibrium (a general chemistry term) between propagating (living) and nonpropagating (nonliving) polymer chains in a polymerization system without permanent chain breaking reactions, it is better to call such polymerization as quasiliving polymerization (see for instance the following reference: Ivan, B., *Macromol. Chem. Phys.* 201, 2621-2628 (2000)). As a matter of fact, the "quasi" phenomena are very important in sciences and engineering, which is indicated also by the fact that the Nobel Prize in Chemistry was awarded to Dan Shechtman for quasicrystals in 2011.

On the basis of the above considerations, the authors are suggested to delete the misleading "reversible-deactivation radical polymerization" term in their manuscript, and use the "quasiliving radical polymerization" term instead.

Reviewer #3 (Remarks to the Author):

The authors have addressed the concerns that I had with the original manuscript, most notably in showing that their assay is viable in a realistic matrix of human blood. They have also addressed my concerns about the applicability of the approach by developing an approach using manual separation by size exclusion chromatography. There are of course further steps required to establish the real potential of this assay. These are both an engineering challenge and require trials in a clinical environment which go beyond the scope of the current study. The novelty of the highly sensitive catalytic assay developed by these authors in my opinion justifies publication of this work.

Please, find a point-by-point answer to the comments of the reviewers in blue. The changes in the main text have been highlighted in yellow.

Reviewer #1 (Remarks to the Author):

The new experiments that the authors have conducted with whole blood spiked with cultured *P. falciparum* strengthen the work considerably. I recommend the following revisions (no further experiments) prior to publication:

1. At this stage of development, it is more appropriate to compare this new method with other laboratory-based tests than with rapid detection test (RDTs, e.g. lateral flow).

I have great appreciation for the advances the authors have made since the initial submission. However, a great deal of work (including conceptual advances to get around trade-offs-- performance vs. cost, time, equipment, skilled labor, etc.) still remain before this test will be comparable in simplicity to RDTs.

The hand-driven chromatography approach quoted below still involves many steps, a lot of time, and substantial laboratory equipment and technical expertise:

"Parasite isolation by size exclusion chromatography: 10 mL of human blood were collected in heparin-coated tubes (Sarstedt Monovette Li-Heparin LH/9 mL). 50 μ L of this sample were transferred into a 1.5 mL Eppendorf tube (polypropylene) and spiked with known quantities of a nonsynchronized *P. falciparum* culture (concentration of parasites determined by Giemsa staining). The RBC membranes were selectively lysed with 250 μ L of a 0.3 % (w/v) solution of saponin in PBS. The lysate was submitted to a size exclusion column consisting of Sepharose 4B CL packed in a Biotage SNAP Ultra 10 g column as the stationary phase with PBS as the mobile phase. Prior to the chromatography, the columns were equilibrated by substituting the 20:80 ethanol/water solution in which the stationary phase is delivered with 50 mL of ultrapure water and then 50 mL of PBS buffer. Then, the lysate was loaded to the column and eluted by passing 6 mL PBS through the column by means of a hand-driven syringe., The first 5 mL of PBS were collected and contained the parasites. A clear band for the presence of hemoglobin was observed and remained on the column, which was afterwards discarded. The isolate was centrifuged in a 15 mL Falcon tube at 5000 x g for 10 min forming a pellet. The supernatant was discarded and the pellet was dried overnight in a vacuum oven at 60 °C. The parasites were further lysed and their contents solubilized in 55 μ L of a 0.4 M NaOH aqueous solution and sonicated for 30 min (sonicator bath Sonoswiss SW3). The sample was then centrifuged for 10 min at 5000 x g to spin down the sample and 50 μ L were introduced into the polymerization assay."

I agree with the authors' enthusiasm for making their new test competitive with RDTs, but that future work is non-trivial. At present, it is more appropriate to compare with laboratory-based tests.

While I don't think further experimental work is required prior to publication, the writing/framing of the current accomplishment should be revised prior to publication.

We thank Reviewer 1 for this insightful comment. The manuscript reports a novel assay for hemozoin detection and focuses mainly on the chemistry of this assay. It is true that at this stage the test still needs to be improved and further engineered to achieve a POC test that is comparable in its operational simplicity with an MRDT. In the current incarnation, the isolation of hemozoin or malaria parasite from blood requires lab-based steps. However, they are not dependent on sophisticated lab equipment and could be carried out in laboratories in resource limited settings. However, it is obvious that significant development efforts are needed in terms of device engineering to integrate sample preparation and the actual assay into a handheld POC diagnostic device.

We removed the comparison of our test to MRDTs from the abstract and we have clearly stated that further improvements are necessary to outperform MRDTs in the conclusions. Our research group is currently investing resources on solving these limitations. We have highlighted the changes in yellow in the text.

2. Parasites μL^{-1} and infected red blood cells μL^{-1} are both used in various places in the manuscript. A clear statement on how to relate these quantities to one another and to ng/mL hemozoin would be helpful in the manuscript (the earlier, the better).

We agree with this accurate comment. The concentration of parasites expressed as infected red blood cells (iRBC) μL^{-1} corresponds to the number of red blood cells that contain at least one parasite in a volume of 1 μL of blood. Instead, the concentration of parasites expressed as parasites μL^{-1} corresponds to the number of parasites contained in a volume of 1 μL of blood. In infections of malaria species such as *Plasmodium falciparum*, there can be cases in which one red blood cell can be coinfecting by more than one parasite. As all RBCs at the same stage of maturity should contain on average the same concentration of haemoglobin, the quantity of haemoglobin that is converted to hemozoin should be similar whether the one RBC is infected by one or more parasites. Thus, we have opted by expressing all parasite concentrations as iRBC μL^{-1} . We have highlighted the changes in yellow in the text.

As it requires extensive calculations to correlate the concentration of hemozoin to the concentration of parasites in iRBC μL^{-1} , they are described and discussed in the SI. We have forwarded the reader to the calculations to the SI where we explain how the concentration of hemozoin corresponds to the iRBC μL^{-1} . The changes are highlighted in yellow in the text.

3. These competing methods:

"Physical methods such as laser desorption mass spectrometry,²¹ Raman spectroscopy,²² laser-induced nanobubble formation,²⁰ multiple-angle polarization scatter separation,²³ or magnetically induced dichroism²⁴⁻²⁶ have been used to detect hemozoin, also in peripheral blood. But none of these methods are used widely due to their technical complexity and/or lack of sensitivity."

could be discussed in a bit more depth, including efforts to make user-friendly, field portable sensors/devices.

It is not necessary to cite these particular studies, but they are offered as examples of the kinds of efforts that should not be dismissed without thoughtful consideration:

<https://pubs.acs.org/doi/10.1021/acssensors.8b00269>

<https://www.nature.com/articles/nm.3622>

This is a good thought as it also demonstrates that hemozoin is a valid biomarker for malaria diagnosis. We have edited this part in the main text and it can be found highlighted in yellow.

Reviewer #2 (Remarks to the Author):

The authors have carried out detailed revision of their manuscript on the basis of the reviewers' comments. It is recommended to be accepted for publication after some minor revision on the basis of the comment below.

As a result of Comment 2 by Reviewer 2, the authors use the term of "reversible-deactivation radical polymerization" instead of the sloppy undefined "controlled/living" term for polymerizations catalyzed by heme containing enzymes or hemin in their revised manuscript. The term "reversible-deactivation polymerization" is even worse than "controlled/living polymerization". As described in every polymer chemistry textbook, chain polymerization, including radical polymerizations as well, proceeds by propagation, and never by "reversible-deactivation". When there is an equilibrium (a general chemistry term) between propagating (living) and nonpropagating (nonliving) polymer chains in a polymerization system without permanent chain breaking reactions, it is better to call such polymerization as quasiling polymerization (see for instance the following reference: Ivan, B., Macromol. Chem. Phys. 201, 2621-2628 (2000)). As a matter of fact, the "quasi" phenomena are very important in sciences and engineering, which is indicated also by the fact that the Nobel Prize in Chemistry was awarded to Dan Shechtman for quasicrystals in 2011.

On the basis of the above considerations, the authors are suggested to delete the misleading "reversible-deactivation radical polymerization" term in their manuscript, and use the "quasiling radical polymerization" term instead.

We thank reviewer 2 for this comment. Though, the term "quasiling radical polymerization" has been extensively used in the past literature, it was discouraged by IUPAC because the reactions are not *per se* a living processes due to the inevitable termination reactions between two radicals. The term "controlled radical polymerization (CRP)" is permitted by IUPAC, but it is recommended to use the term "reversible-deactivation radical polymerization". Therefore, we will stick to this nomenclature in the text. Please see <https://www.degruyter.com/view/IUPAC/iupac.82.0001> for the definition by IUPAC.

Reviewer #3 (Remarks to the Author):

The authors have addressed the concerns that I had with the original manuscript, most notably in showing that their assay is viable in a realistic matrix of human blood. They have also addressed my concerns about the applicability of the approach by developing an approach using manual separation by size exclusion chromatography. There are of course further steps required to establish the real potential of this assay. These are both an engineering challenge and require trials in a clinical environment which go beyond the scope of the current study. The novelty of the highly sensitive catalytic assay developed by these authors in my opinion justifies publication of this work.

We thank reviewer 3 for this kind comment and we fully agree with the state of our research. In this work we show the fundamentals of the amplification of the malaria biomarker hemozoin via precipitation polymerisation. We have shown that this is a potential approach to diagnose malaria, but we agree that a lot of engineering and patient testing is needed to achieve a real diagnostic test. Therefore, we are investing research resources towards this goal.